chemical engineering/fluid mechanics

mass transfer, rising behaviour, coalescence behaviour, cutting behaviour, concentration distribution

**Author for correspondence:**
Guanghui Chen
e-mail: guanghui@qust.edu.cn

This article has been edited by the Royal Society of Chemistry, including the commissioning, peer review process and editorial aspects up to the point of acceptance.

# Interactions between gas–liquid mass transfer and bubble behaviours

Xin Li[1], Weiwen Wang[1], Pan Zhang[2], Jianlong Li[1]
and Guanghui Chen[1]

[1]College of Chemical Engineering, Qingdao University of Science and Technology, Qingdao 266042, People's Republic of China
[2]College of Electromechanical Engineering, Qingdao University of Science and Technology, Qingdao 266061, People's Republic of China

PZ, 0000-0003-3344-4823; GC, 0000-0002-3459-4834

Interactions between gas–liquid mass transfer and bubble behaviours were investigated to improve the understanding of the relationship between the two sides. The $CO_2/N_2$-water system was applied to study the bubble behaviours based on the volume-of-fluid (VOF) model. The mass transfer conditions were taken into consideration when the fluid field was analysed. The bubble behaviours were compared with and without mass transfer. The results show that the absolute slopes of the curves for mass fraction inside the single rising bubbles, with diameters from 3 to 6 mm, decrease from 0.09325 to 0.02818. It means that small single bubbles have higher mass transfer efficiency. The daughter bubbles of cutting behaviour and initial side-by-side bubbles of coalescence behaviour also perform better than the initial large bubbles and coalesced bubbles, respectively. The bubble behaviours affect the mass transfer process. However, the latter also reacts upon the former. The critical intervals between the side-by-side bubbles decrease from 2.0 to 0.9 mm when the bubble diameter changes from 3 to 7 mm. For the coalescence behaviour without mass transfer, the critical intervals are larger because there is no influence of concentration around the bubbles on the bubble motion. The coalescence of cut daughter bubbles is also influenced by the concentration. It was suggested that the interaction between the gas–liquid mass transfer and bubble behaviours cannot be ignored.

## 1. Introduction

Gas–liquid multi-phase flow is a common phenomenon in chemical and biochemical processes [1–4]. The behaviours of gas phase, which is shown as bubbles in the liquids, are quite

complex. Usually, the bubble rising process is accompanied by mass transfer, which further increases the complexity of bubble behaviours.

The previous studies have been conducted to investigate the bubbles coalescence and break-up behaviours. The coalescence behaviour can be classified into: in-line [5–7] and side-by-side [8,9]. For the in-line bubbles coalescence, there are two coaxial bubbles and the upper one is called the leading bubble, while the other one is called the trailing bubble. The drag force of the liquid acting on the bubbles [10] and the interaction between the in-line bubbles [11] were studied. The coalescence behaviours of the in-line bubbles are not only direct coalescence but also coalescence with conjunction [12]. For the side-by-side bubble coalescence, there are two parallel bubbles rising together initially and then the interval between them decreases until they coalesce [13]. The properties of the gas and the liquid have impacts on the coalescence [14]. Just like the in-line bubbles, the coalescence behaviours of the side-by-side bubbles can also be classified into two types. One of them is direct coalescence and the other one is coalescence with bouncing [15]. Moreover, the possibility of coalescence is determined by the initial interval between the bubbles and the coalesced bubbles may also break up if the diameter is large enough [16].

Bubble break-up behaviour also usually occurs in the rising process [17–19]. It is generally recognized that the bubbles tend to break up when the surface tension of the interface is changed greatly. There is a special bubble break-up condition called cutting, in which the bubble is cut by a wire or wire mesh set along their rising path [20,21]. When the bubbles contact the wire, three behaviours of cut, bypass and stick will happen. The behaviours are decided by the rising velocity and properties of gas–liquid system. When the wire mesh is set, the bubbles will contact the centre or the crossing of the meshes. Similar behaviours would occur. In our previous study [22], a wire mesh was set above the tray in the distillation column, and the results showed that the mean bubble diameter decreased sharply. The above studies are of great guiding significance to the coalescence and break-up behaviours, but the influences of the behaviours on mass transfer have not been investigated and the interaction between them is also rarely investigated.

The interaction between the bubble behaviours and the gas–liquid mass transfer for a single bubble rising process has been recognized. However, the researchers paid more attention to liquid-side mass transfer, while the gas-side or coupled two-side mass transfer is rarely studied [23–26]. The liquid-side mass transfer process is influenced by the properties of the liquid such as density, surface tension and so on. The volume-of-fluid (VOF) method is widely used in the simulation studies [27]. For the calculation of the mass transfer process, a two-variable method is investigated and shows good computations for the high Schmidt and Reynolds number bubbles [28]. Direct numerical simulation (DNS) could also simulate the mass transfer process. The effect of interface contamination on mass transfer can be calculated accurately based on the method [29]. Experimental investigation about mass transfer processes is another research focus [4]. A new experimental method with some specific chromogenic methods was developed [30]. There are several advantages of the new method such as the visualization of the concentration distribution. Besides studying the liquid-side mass transfer, Saboni *et al.* [31] studied the soluble substance transportation from liquid into bubble and the concentration distribution forms in bubble based on simulation. However, the gas-side mass transfer process and the behaviours with mass transfer were rarely studied.

Based on the summary about the previous works, the coalescence and break-up behaviours for bubble rising with mass transfer need to be further investigated for improving chemical equipment performance. Thus, the objective of this study is to reveal the states of mass transfer under the two conditions. For that, different dimensional bubbles with the composition of 80 wt% $CO_2$ and 20 wt% $N_2$ were selected as the study subjects. The characteristics of the behaviours with mass transfer, which were rarely reported, were analysed to develop a deeper understanding of the transport processes (table 1). The mass transfer of single bubbles with different diameters and the relationship between the concentration wakes and fluid field were investigated. The bubble cutting behaviour and the interactions between mass transfer and bubble motion were revealed. Moreover, the state of mass transfer and the fluid field for side-by-side bubbles were studied. An important parameter of the critical interval was also analysed to find out the differences in the behaviour with and without mass transfer.

# 2. Model set-up

The rising bubble behaviours with mass transfer were simulated based on the multi-phase flow model. The gas phase is the mixture of $CO_2$ and $N_2$, while the liquid phase is water. The solubility of $CO_2$ is

**Table 1.** Nomenclature.

| | |
|---|---|
| $d_b$ | bubble diameter, m |
| $D$ | diffusion coefficient, $m^2\,s^{-1}$ |
| $F_b$ | buoyancy force |
| $F_G$ | gravity force |
| $F_s$ | surface tension source term |
| $F_v$ | volume fraction |
| $F_y$ | total force |
| $g$ | gravity acceleration magnitude, $m\,s^{-2}$ |
| $J$ | mass flux, $g\,cm^{-2}\,s^{-1}$ |
| $m$ | dissolved $CO_2$, kg |
| $\mathbf{n}$ | unit vector |
| $N$ | mass flux, $kg\,m^{-3}\,s^{-1}$ |
| $P$ | pressure, Pa |
| $t$ | time, s |
| $\mathbf{u}$ | velocity, $m\,s^{-1}$ |
| greek symbols | |
| $\alpha$ | volume fraction |
| $\sigma$ | surface tension, $N\,m^{-1}$ |
| $\mu$ | viscosity, Pa s |
| $\rho$ | density, $kg\,m^{-3}$ |
| $\kappa$ | interface curvature |

**Table 2.** Properties of the gas and liquid substances.

| material | density ($kg\,m^{-3}$) | viscosity ($kg\,m^{-1}\,s^{-1}$) | surface tension ($N\,m^{-1}$) |
|---|---|---|---|
| water | 998.2 | $1.003 \times 10^{-3}$ | 0.07196 |
| $CO_2$ | 1.7878 | $1.37 \times 10^{-5}$ | — |
| $N_2$ | 1.138 | $1.663 \times 10^{-5}$ | — |

0.169 g/100 g water, while the solubility of $N_2$ is only $1.89 \times 10^{-3}$ g/100 g water. The properties of the materials are shown in table 2.

## 2.1. Concept model

The mechanism of mass transfer is shown in figure 1. First, the $CO_2$ diffuses to the gas–liquid interface from the inner gas. Then in the interface, the $CO_2$ of the gas phase is in equilibrium with that of the liquid phase. The mass flux through the interface is continuous. In other words, the amount of $CO_2$ transferred from the inner gas to the interface is equal to that transferred from the interface to the liquid. Finally, the $CO_2$ diffuses in the liquid phase.

## 2.2. Hypothesis

The simulations were implemented according to the following assumptions:

(1) The gas and liquid fluids are regarded as incompressible.
(2) The properties of gas and liquid fluids are pressure-independent such as density, viscosity and surface tension.

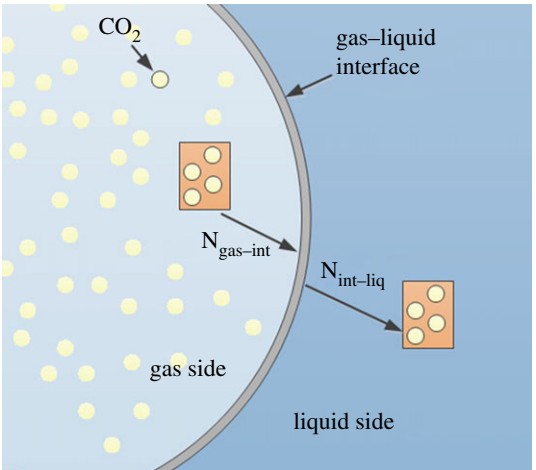

**Figure 1.** Mechanism of $CO_2$ mass transfer processes.

(3) Initial spherical bubbles are set at the bottom and they are driven by buoyancy so that the diameters of the bubbles and the intervals between the side-by-side bubbles can be controlled accurately.

(4) Only the $CO_2$ can transfer from gas side to liquid side, and $N_2$ is assumed to be not soluble in water. This assumption is reasonable by comparing the solubility of the two gas substances.

## 2.3. Numerical model

Both the gas and liquid phases obey the continuous equation and the Navier–Stokes equation as follows:

$$\nabla \cdot u = 0 \tag{2.1}$$

and

$$\frac{\partial(\rho\mathbf{u})}{\partial t} + \nabla \cdot (\rho\mathbf{u}\mathbf{u}) = -\nabla p + \nabla \cdot [\mu(\nabla\mathbf{u} + \nabla\mathbf{u}^T)] + \rho\mathbf{g} + F_S, \tag{2.2}$$

where $u$ is the velocity vector, $\rho$ is the density, $p$ is the pressure, $\mu$ is the viscosity, and $F_S$ represents surface tension source term, which is taken into account by the continuum surface force (CSF) model and can be expressed by

$$F_S = \sigma \frac{\rho\kappa\nabla\alpha}{0.5(\rho_1 + \rho_2)}, \tag{2.3}$$

where $\sigma$ is the surface tension, $\kappa = \nabla \cdot \hat{n}$, $\hat{n} = n/|n|$, $n = \nabla\alpha_q$.

Based on the volume fractions $F_V$, the density and viscosity of the mixture are calculated by

$$\rho = F_{V_1}\rho_1 + (1 - F_{V_1})\rho_2 \tag{2.4}$$

and

$$\mu = F_{V_1}\mu_1 + (1 - F_{V_1})\mu_2. \tag{2.5}$$

The $CO_2$ mass transfer processes are expressed by species transport equations as follows:

$$\frac{\partial(\rho_g Y_{i,g})}{\partial t} + \nabla \cdot (\rho_g \mathbf{u} Y_{i,g}) = \nabla \cdot \rho_g D_{i,mg} \nabla Y_{i,g}, \tag{2.6}$$

where $Y_{i,g}$ is the $CO_2$ mass fraction in the gas mixture, $D_{i,mg}$ is the diffusion coefficient of $CO_2$–$N_2$ mixture with a value of $1.67 \times 10^{-5}\,\mathrm{m^2\,s^{-1}}$.

For liquid phase, the $CO_2$ transports from the interface and then diffuses inside the liquid. The equation is shown as follows:

$$\frac{\partial(\rho_l Y_{i,l})}{\partial t} + \nabla \cdot (\rho_l \mathbf{u} Y_{i,l}) = \nabla \cdot \rho_l D_{i,ml} \nabla Y_{i,l}, \tag{2.7}$$

where $Y_{i,l}$ is the $CO_2$ mass fraction in the liquid, $D_{i,ml}$ is the diffusion coefficient of $CO_2$ in the water with a value of $1.96 \times 10^{-9}\,\mathrm{m^2\,s^{-1}}$. The mass flux of $CO_2$ transferring through the interface is shown as below.

$$N_{\mathrm{gas-int}} = -D_{i,mg}\frac{\partial C_{i,g}}{\partial Z} = N_{\mathrm{int-liq}} = -D_{i,ml}\frac{\partial C_{i,l}}{\partial Z}, \tag{2.8}$$

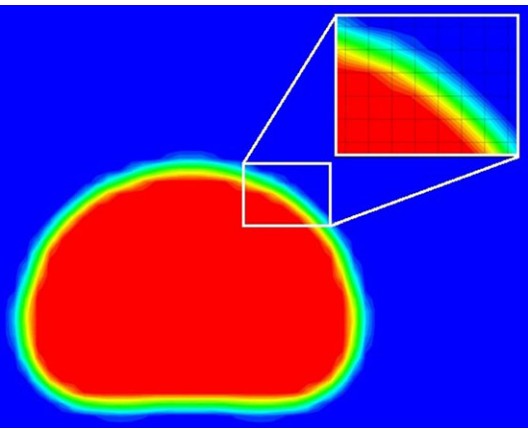

**Figure 2.** Grids of the interface.

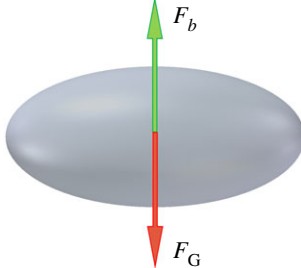

**Figure 3.** Force analysis for bubbles.

where $N_{int-liq}$ is the mass flux from the interface to liquid phase and $N_{gas-int}$ is the mass flux from gas phase to interface. Based on the VOF model, the grids of the interface whose phase volume fraction is in the range of 0 to 1 are searched and the $CO_2$ transfer process is implemented on the interfacial grids (figure 2).

Initially, the bubbles were driven by buoyancy and patched at the bottom of the domain, which is filled with quiescent water. The buoyancy force and gravity force act on the bubble during the whole rising process (figure 3). The total force can be calculated by equation (2.9). The two sides and the bottom of the domain were set as no-slip wall boundary condition, while the top was set as pressure outlet with backflow of water.

$$F_y = F_b - F_G. \tag{2.9}$$

## 2.4. Geometry and solution methods

The computation domain was set as a rectangle with 80 mm in width and 80 mm in height. The grid of the two-dimensional geometry model was meshed to two different types, including $0.2 \times 0.1$ mm (320 000) for the bubble cutting and $0.2 \times 0.2$ mm (160 000) for the single bubble rise and bubble coalescence. A grid independence check was applied to ensure the refined mesh does not have influence on the results. For the single bubble rise and bubble coalescence, three meshes of 100 000, 160 000 and 200 000 grids were compared based on the dissolved $CO_2$ in water. For the bubble cutting behaviour, three meshes with 240 000, 320 000 and 400 000 grids were compared. Figure 4 shows the compared results about the three meshes, and it can be found that the dissolved $CO_2$ of the mesh with 160 000 grids is similar to that with 200 000 grids for single bubble rises and bubble coalescence with maximum deviations of 7.73% and 4.9%, respectively. While for the bubble cutting, the mesh with 320 000 grids is similar to that with 400 000 grids and the maximum deviation is 4.3%. Therefore, the selected grid numbers are large enough to obtain accurate results. Under the bubble cutting condition, the tenuous wire was set at 40 mm in height from the bottom of the domain. The shape of the wire is square so that high-quality quadrilateral grids can be meshed. Its side length is only 0.4 mm which is much smaller than the bubble diameters so the influence of wire shape can be ignored. The walls of the wire were set as wall boundary conditions.

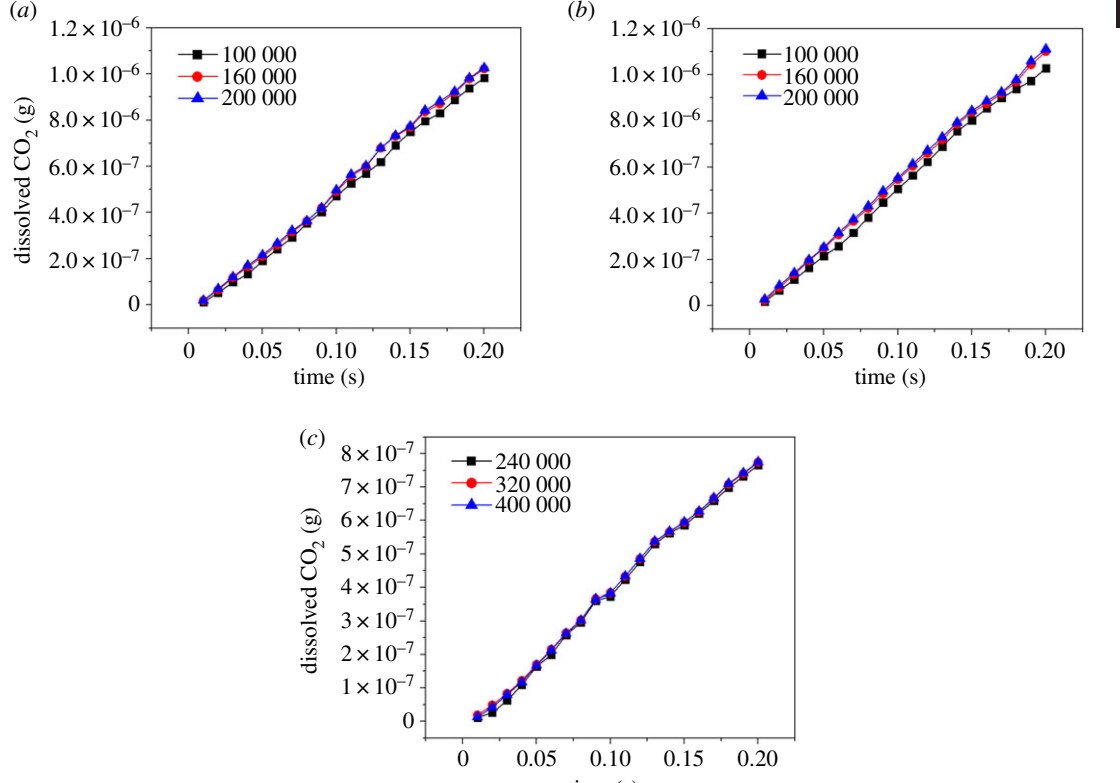

**Figure 4.** Compared results of dissolved $CO_2$ for different meshes: (a) 5 mm single bubble rise; (b) 3 mm side-by-side bubble coalescence and (c) 5 mm bubble cutting.

The VOF multi-phase model was applied to simulate the gas–liquid flow. A transient pressure-based solver was applied and the volume fraction equation was solved by using the explicit Geo-Reconstruct scheme. The pressure–velocity coupling method was the SIMPLE scheme and the flow equations were discretized by the QUICK scheme. The PRESTO! scheme was used to discretize pressure and the first-order upwind scheme was used for $CO_2$ mass fraction equation discretization. The iteration time step was set as $1.0 \times 10^{-4}$ s and the max iteration number of each time step was 20.

## 2.5. Model validation

The CFD model was verified by comparing the experimental results from Zahedi et al. [32] and Kong et al. [33]. It should be noted that though three-dimensional experiments were conducted, the bubble photos were taken with a two-dimensional perspective so that they are suitable to be used to validate the model. The computation domain of $80 \times 100$ mm was used. In the work of Zahedi et al., the air was injected through the orifice with 4.5 mm diameter and water was initially filled in the domain. The air flow rate was controlled at $2.5 \times 10^{-7}$ $m^3 s^{-1}$ by a pump with a flow controller. The formation and rising processes of the bubble were recorded by a high-speed camera with the speed of 60 frames $s^{-1}$. The processes were analysed by the recorded movie files with snapshots taken every 0.016 s. Figure 5 shows the comparisons of the simulated bubble formation and rising motion with the experiment. As shown in figure 5, the shapes of simulated bubbles are similar to those of measured bubbles. The final rising velocity is also compared with each other as shown in table 3. The relative error is only 4% which means that the results calculated by the model agree with those of the experiment well. Besides, the model was also validated based on the concentration wake structure of the dissoluble gas in the liquid by comparing the results obtained from Kong et al. [33]. To obtain the concentration wake structure, dual-emission dye C-SNARF-4F was selected and the laser-induced fluorescence (LIF) technique was applied. The pH distribution around the bubble was captured and then the concentration distribution could be calculated by the conversion function. The computation domain of $80 \times 80$ mm was used. A dissolving carbon dioxide bubble rises in the water. The left images in figure 6 show that the concentration wake structure below the bubbles is different in the

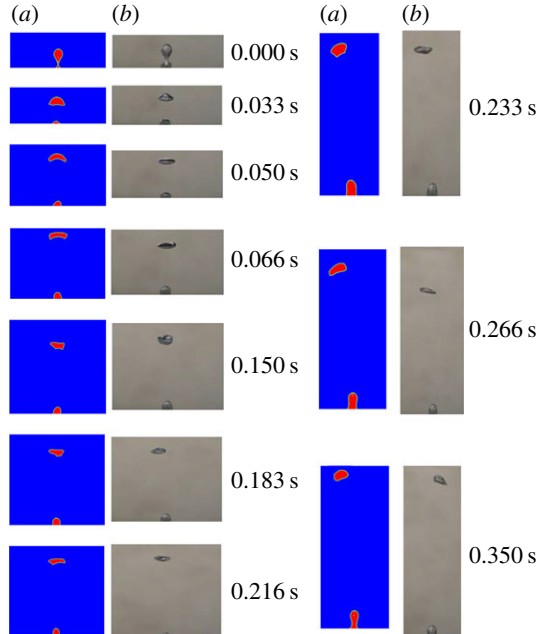

**Figure 5.** Comparison of bubble formation and rising motions for (*a*) simulation and (*b*) experiment [32].

**Table 3.** Comparison of rising velocity for measured bubble [32] and simulated bubble.

|  | rising velocity (m s$^{-1}$) |
| --- | --- |
| measured bubble | 0.224 |
| simulated bubble | 0.234 |
| relative error (%) | 4 |

experiment. The right images are the results of simulations. In each pair of subfigures, the calculated high concentration zones below the bubbles are the same as the zones captured in the experiments. The outlines of the concentration wake structures in the experiment and simulation are also similar. The results show that the wake structure calculated by the model accords well with the experimental observation, thus the model is validated.

# 3. Results and discussion

## 3.1. Single bubble behaviours with mass transfer

Single bubbles with different diameters of 2, 3, 4, 5 and 6 mm, were studied. The dissolved $CO_2$ mass fraction of the single bubbles in water is shown in figure 7. The $CO_2$ mass fraction distribution can reflect the rising trajectory indirectly. The bubbles with the diameters of 2 and 3 mm rise unsteadily, while larger bubbles rise vertically in the computation domain. The unsteady path can increase the residence time of the small bubbles and then extend the mass transfer process. For example, the residence time of 3 mm bubble is longer than that of 4 mm bubble and the dissolved $CO_2$ in water is $1.60 \times 10^{-9}$ kg for 3 mm bubble while it is $1.52 \times 10^{-9}$ kg for 4 mm bubble. There are concentration wakes below the bubbles. Take the 5 mm bubble for instance (figure 7d), three pairs of obvious symmetric wakes occurred. Through analysing the bubble deformation during the rising process, it can be seen that the symmetric concentration wakes are related to the bubble shapes (figure 8). Figure 8 shows the history of 5 mm bubble deformation under the background of $CO_2$ mass fraction. Every time when the bubble becomes flat, the deformation of bubble shape changes the flow patterns, and the dissolved $CO_2$ diffuses with the liquid flowing. The mentioned relationship among bubble deformation, flow patterns and $CO_2$ diffusion can be verified in figure 8. The flow vortexes which are

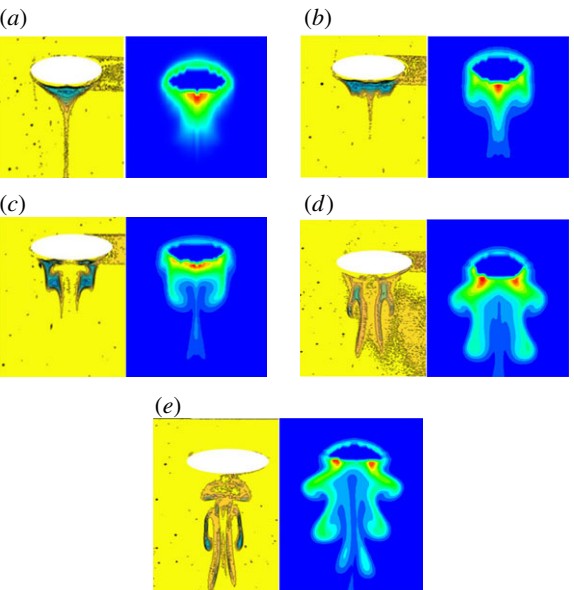

**Figure 6.** Comparison of wake structure for experiment [33] and simulation: (*a*) 40 ms; (*b*) 60 ms; (*c*) 90 ms; (*d*) 110 ms and (*e*) 120 ms.

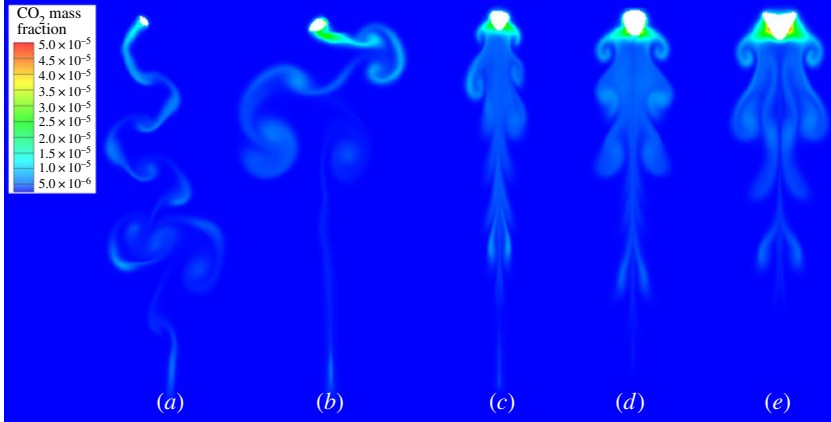

**Figure 7.** $CO_2$ mass fraction distribution in water for different bubbles: (*a*) 2 mm; (*b*) 3 mm; (*c*) 4 mm; (*d*) 5 mm and (*e*) 6 mm.

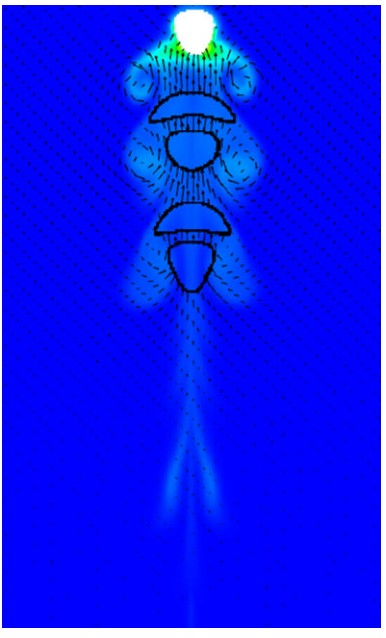

**Figure 8.** History of 5 mm bubble deformation under the background of $CO_2$ mass fraction.

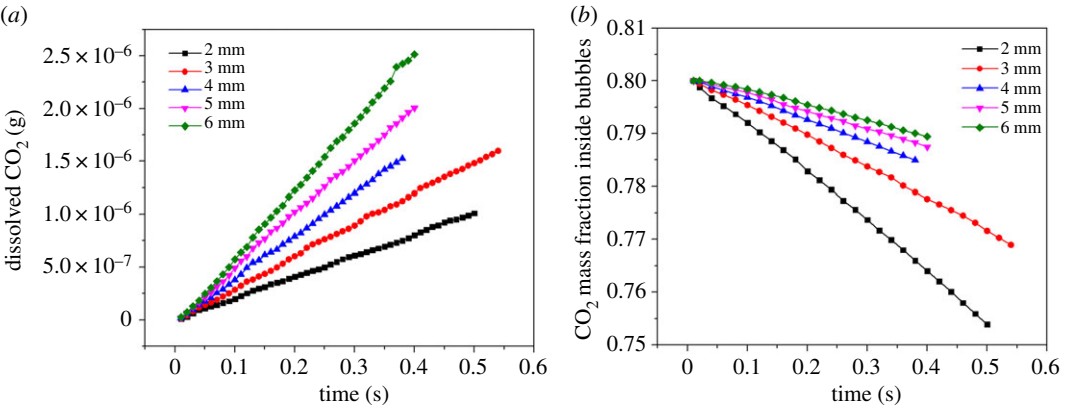

**Figure 9.** State of mass transfer for different bubbles: (*a*) dissolved $CO_2$ in water and (*b*) change of $CO_2$ mass fraction inside bubbles.

**Table 4.** Global mass flux of single bubbles with different diameters.

| bubbles with different diameters (mm) | d$m$/d$t$ | global mass flux (g cm$^{-2}$ s$^{-1}$) |
|---|---|---|
| 2 | $2.013 \times 10^{-9}$ | $1.603 \times 10^{-5}$ |
| 3 | $3.005 \times 10^{-9}$ | $1.063 \times 10^{-5}$ |
| 4 | $4.091 \times 10^{-9}$ | $8.143 \times 10^{-6}$ |
| 5 | $5.100 \times 10^{-9}$ | $6.497 \times 10^{-6}$ |
| 6 | $6.488 \times 10^{-9}$ | $5.739 \times 10^{-6}$ |

shown by vectors occurred at the same places where concentration wakes locate. The global mass flux of the single bubbles is calculated by the following equation [34]:

$$J \cdot \pi d_b^2 = \frac{\mathrm{d}m}{\mathrm{d}t}, \tag{3.1}$$

where d$m$/d$t$ is the slope of the dissolved $CO_2$ curve.

Table 4 shows the calculated global mass flux of different single bubbles. The mass flux becomes lower when the bubbles become larger and it shows that the mass transfer efficiency of initial small single bubbles is better than that of larger bubbles. The relationship between the bubble diameter and mass flux was correlated and the correlation is written as follows:

$$J = 4 \times 10^{-8} \cdot d_b^{-0.947}, \tag{3.2}$$

where the *R*-square value is 0.9976, which means that the correlation has a high accuracy.

Figure 9*a* shows the total amount of $CO_2$ in water accumulated with time. If the residence time of bubbles with different diameters is the same, more $CO_2$ from larger bubbles transports into the water due to the larger interface area. It should be noted that the final amount of dissolved $CO_2$ for 3 mm bubble is higher than that of 4 mm bubble because of the longer residence time. Figure 9*b* shows the change of $CO_2$ mass fraction inside the bubbles. The variation range increases with the decrease in bubble diameter. Compared with the curves in figure 9*a*, more $CO_2$ in larger bubbles transports into water while the residual $CO_2$ inside the bubbles is still higher (figure 9*b*). Besides, the absolute slopes of the curves for mass fraction inside the single rising bubbles with the diameters from 3 to 6 mm decrease from 0.09325 to 0.02818 in figure 9*b*. Hence, based on the above analysis of $CO_2$ mass fraction on the gas side and dissolved $CO_2$ on the liquid side, it is verified that small single bubbles have higher mass transfer efficiency though the interface areas are small.

## 3.2. Bubble cutting behaviours with mass transfer

Generally, small bubbles could intensify the mass transfer. Hence, several methods to reduce the bubble diameter have been studied [35–37]. Cutting the large bubbles to smaller ones by a wire or wire mesh

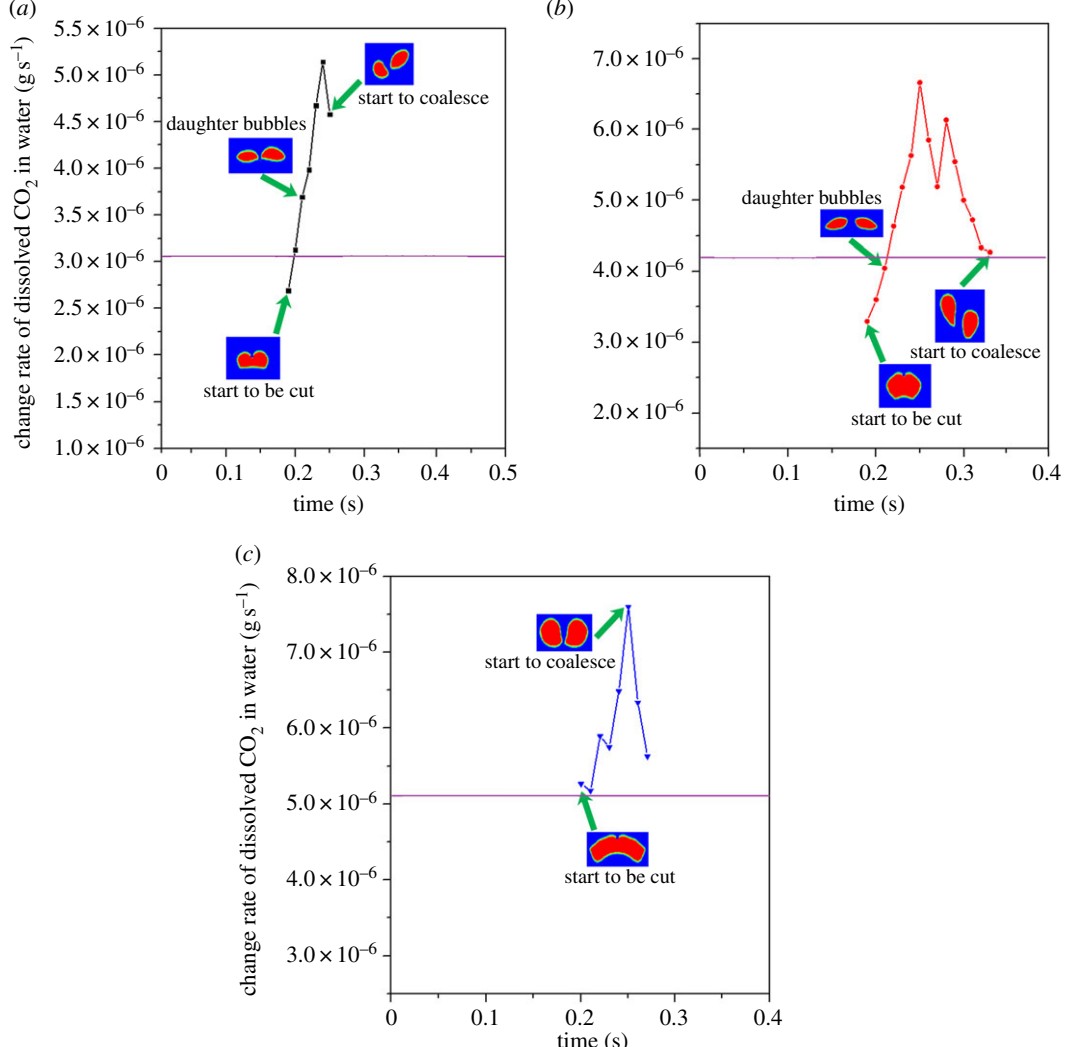

**Figure 10.** Change rate of dissolved $CO_2$ in water under different bubble cutting conditions: (a) 4 mm; (b) 5 mm and (c) 6 mm.

which is set on the rising path is a convenient and cheap method compared with designing new trays or gas distributors.

Bubbles with initial diameters of 4, 5 and 6 mm were investigated when they were cut by a wire. Figure 10 shows the change rate of dissolved $CO_2$ in water. The change rate can reflect the mass transfer efficiency at different steps. The bubbles rise vertically and then contact the wire at 0.2 s. The 4 and 5 mm single bubbles are completely cut into two daughter bubbles at 0.22 s. For 6 mm single bubble, the cut process finishes at 0.25 s. Then the daughter bubbles rise continuously. It can be seen in figure 10 that the change rate of dissolved $CO_2$ in water increases dramatically after the single bubbles are being cut. There is an obvious peak value of the change rate for each bubble with different initial diameter and they are much higher than the average value shown as the purple lines. During the rising process of the daughter bubbles, they collide with each other. Then they coalesce and the change rate of dissolved $CO_2$ decreases obviously. This is because the interval between them is not large enough and the relevant analysis would be implemented later. When the bubbles coalesce, the change rate decreases obviously. The final masses of the dissolved $CO_2$ in water are $1.89 \times 10^{-9}$ kg for 4 mm bubble, $2.09 \times 10^{-9}$ kg for 5 mm bubble and $2.13 \times 10^{-9}$ kg for 6 mm bubble. The $CO_2$ mass fraction inside the 5 mm diameter bubble is shown in figure 11. The $CO_2$ mass fraction decreases from 0.8 to 0.787 over time and the rising processes can be divided into three steps. Firstly, the initial single bubble rises and it was cut by the wire at 0.21 s. The $CO_2$ mass fraction inside the bubble decreases at a stable rate. The slope of the curve in this step is −0.02527. Then, two daughter bubbles occurred and the decrease rate intensified until 0.33 s. The slope becomes −0.03404. Finally, the daughter bubbles coalesce and the slope becomes −0.02474. The slopes of the three steps can reflect the mass transfer conditions. At the first and the third steps, the $CO_2$ transports through single

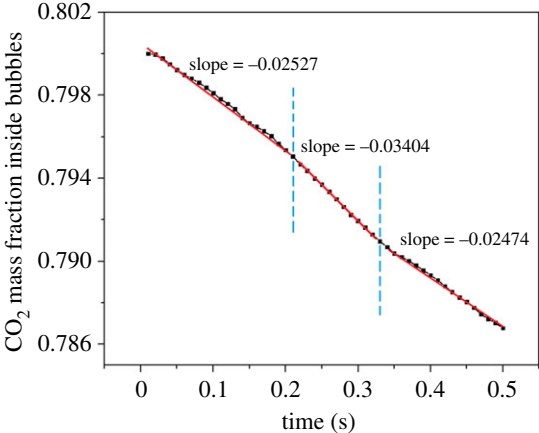

**Figure 11.** Change of $CO_2$ mass fraction inside bubbles under bubble cutting conditions.

bubbles with the similar diameter, so the slopes are similar. While at the second step, two small bubbles rise simultaneously and the mass fraction decreases more quickly. Hence, cutting bubbles can indeed intensify the mass transfer efficiency. However, there is still a problem which has not been explained and that is why the residence time of daughter bubbles is so different when the initial bubble diameters are changed.

Figure 10 exhibits that the daughter bubbles of 4 mm bubble are retained for 0.05 s while those of 5 mm and 6 mm bubbles are retained for 0.12 and 0.02 s, respectively. The moments of the bubbles contact with the wire are shown in figure 12 under the background of $CO_2$ mass fraction in water. When the bubbles are cut by the wire, high mass fraction regions occurred. Huang & Saito [38] speculated that the concentration around the single rising bubble has impacts on bubble motion based on experiments. Hence, under the bubble cutting condition, the $CO_2$ mass fraction distribution would also influence the bubble motion. For the 4 mm bubble, the high concentration region is directly below the bubble after being cut so that the daughter bubbles are pushed up vertically. While for the 5 mm bubble, the daughter bubbles are pushed on a negative direction. Therefore, the daughter bubbles produced by 4 mm bubble have more opportunities to collide compared with those produced by 5 mm bubble. For the 6 mm bubble, the daughter bubbles are pushed on relative direction so they coalesce quickly. To further study the influence of mass transfer on bubble motions, the rising process of 5 mm bubble was investigated (figure 13). The white lines are axel wires. When there is no mass transfer, the bubble rises vertically at the very start and then it deviates from the axel wire. The deviation is caused by the fluid field and there is no mass transfer to offset the deviation. As a result, the bubble is cut into two unequal-diameter daughter bubbles. But for the bubble with mass transfer, it rises along the axel wire and then is cut into two equal-diameter daughter bubbles. When the bubble moves close to the wire, the concentration distribution below the bubble is symmetric (figure 13c). The high concentration regions also arrange symmetrically. Hence, the bubble is pushed vertically and then cut into two equal-diameter daughter bubbles. The comparison shows that the mass transfer cannot be ignored when studying the bubble behaviours.

## 3.3. Coalescence of side-by-side bubbles with mass transfer

As mentioned above, the coalescence behaviours of daughter bubbles produced from large bubbles have been analysed, and the $CO_2$ mass fraction distribution around the rising bubbles influences the bubble motion. Under the condition of the coalescence for daughter bubbles, the mass transfer process has already taken place and the concentration distribution has formed. However, if there are two initial side-by-side bubbles with the same diameters, the coalescence behaviours may be different. The studies about the coalescence of side-by-side bubbles have been reported, but the mass transfer is not taken into consideration, and the interactions of the behaviour and mass transfer were also rarely analysed based on gas side and liquid side.

Initial bubbles with diameters of 3, 5 and 7 mm were investigated. The bubbles were set at the bottom of the domain and there was no $CO_2$ dissolved in the water initially. The interval (L) between the bubbles decides the probabilities to coalesce. Table 5 shows the critical interval for the bubbles to coalesce with

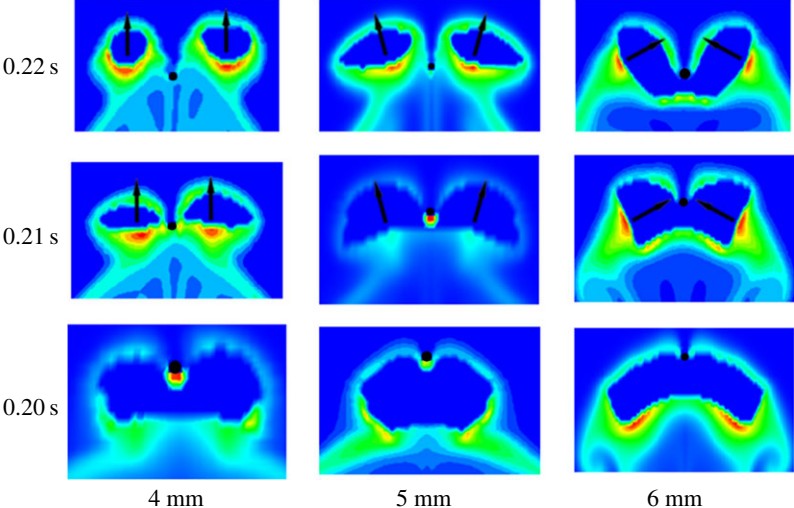

**Figure 12.** Moments of the bubbles contact with wire under the background of $CO_2$ mass fraction in water.

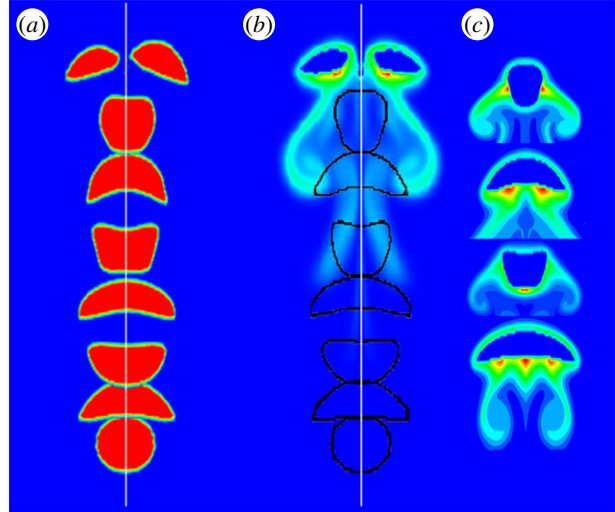

**Figure 13.** Comparison of bubble rising processes: (a) without mass transfer, (b) with mass transfer and (c) concentration distribution of the bubble.

**Table 5.** Critical interval for the side-by-side bubbles to coalesce with and without mass transfer.

| bubble diameter, $D_b$ (mm) | with mass transfer | | without mass transfer | |
|---|---|---|---|---|
| | interval, $L$ (mm) | $L/D_b$ | interval, $L$ (mm) | $L/D_b$ |
| 3 | 2.0 | 0.67 | 2.1 | 0.70 |
| 5 | 1.8 | 0.36 | 2.1 | 0.42 |
| 7 | 0.9 | 0.13 | 1.0 | 0.14 |

and without mass transfer. The interval is the shortest distance between the interfaces of the two bubbles at the same horizontal line, while the critical interval means the longest interval for the bubbles to coalesce. The ratio of the interval to bubble diameter reflects the difficulty of bubble coalescence. For 3 mm bubbles with mass transfer, the ratio is 0.67 while for 5 mm bubbles and 7 mm bubbles, the ratios are 0.36 and 0.13, respectively. It is clear that smaller side-by-side bubbles are easier to coalesce. For example, 3 mm bubble can coalesce while the 5 and 7 mm bubbles cannot coalesce when the interval is 2.0 mm for the bubbles. As for the condition without mass transfer, the ratio of 3 mm

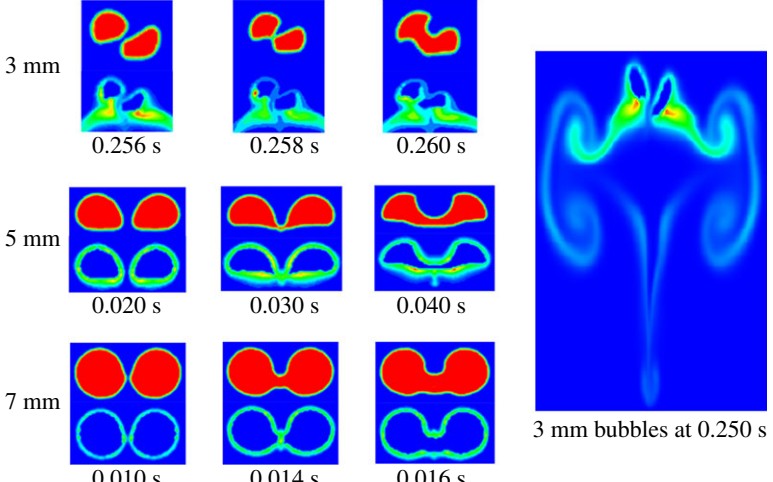

**Figure 14.** Moments of side-by-side bubble coalescence.

bubbles increases to 0.70 and it is 66.7% and five times higher than that for 5 mm bubbles and 7 mm bubbles, respectively. Therefore, the mass transfer also has an impact on the coalescence behaviour for side-by-side bubbles. Moreover, it should be noted that the following studies are all about the behaviour with mass transfer. Figure 14 shows the moments of side-by-side bubble coalescence under critical interval. Each subfigure has two contours. The upper one is the gas volume fraction, and the lower one is the $CO_2$ mass fraction in water. For 5 and 7 mm side-by-side bubbles, they coalesced immediately when they started to rise under critical interval. The $CO_2$ mass fraction distribution around the parallel bubbles is relatively homogeneous under the condition of the critical interval so that the coalescence behaviour is slightly influenced by the concentration distribution. When the bubbles rise, the interfaces located on the two sides of the interval move closer and contact with each other. Then, the gas inside the two bubbles diffuses and the coalesced bubbles are formed. For 3 mm side-by-side bubbles, similar coalescence behaviour can be found when the interval is less than 1.5 mm. Under other interval conditions, the 3 mm side-by-side bubbles rise away from each other initially and then move closer to contact with each other (figure 14).

The surrounding flow patterns would change with the change in the bubble behaviour when the bubbles coalesce. Taking the 5 mm side-by-side bubbles, for example, the coalescence behaviour can be divided into three steps, which are approach step (figure 15a), touch step (figure 15b) and fusion step (figure 15c). At the approach step, the fluid at the interval flows downward and the flow velocity at the interval is higher. The high-velocity area has lower pressure so that the interfaces move closer due to the pressure difference. Besides, two vortexes occurred inside each bubble (see the black frame in figure 15a). At the touch step, the two interfaces contact each other and there is a narrow channel formed between the two bubbles. The two vortexes near the channel move upward and there are two new symmetric vortexes formed below the channel in the water. At the fusion step, the channel becomes wider and the vortexes near the interval disappear. The fluid above the channel flows upward and the reason for the change of flow pattern is related to the bubble deformation. Figure 15d shows the flow pattern around the coalesced bubble. Two vortexes exist on the two sides of the bubble. The channel is not obvious with the gas fusing. At this time, the flow pattern around the coalesced bubbles is similar to that around ordinary single bubbles.

The coalescence behaviour of side-by-side bubbles has effects on the mass transfer processes. The $CO_2$ transported from 3 mm bubbles to water was investigated under the condition of critical interval since the two side-by-side bubbles did not coalesce immediately when they rose. Hence, the mass transfer processes can be compared before and after coalescing. Figure 16 shows the change tendencies of dissolved $CO_2$ and $CO_2$ mass fraction inside bubbles. It should be noted that the side-by-side bubbles coalesce at 0.21 s. In figure 16a, the dissolved $CO_2$ in water increases to $1.10 \times 10^{-9}$ kg from zero before 0.21 s and the slope of the curve is $5.679 \times 10^{-6}$. Then the bubbles coalesce and the dissolution rate becomes lower, which can be reflected by the reduction of the slope ($3.880 \times 10^{-6}$). As for the $CO_2$ mass fraction inside bubbles, the absolute value of slope for side-by-side bubbles is also higher than that for the coalesced bubble. Therefore, the coalescence behaviour of side-by-side bubbles reduces the mass transfer efficiency.

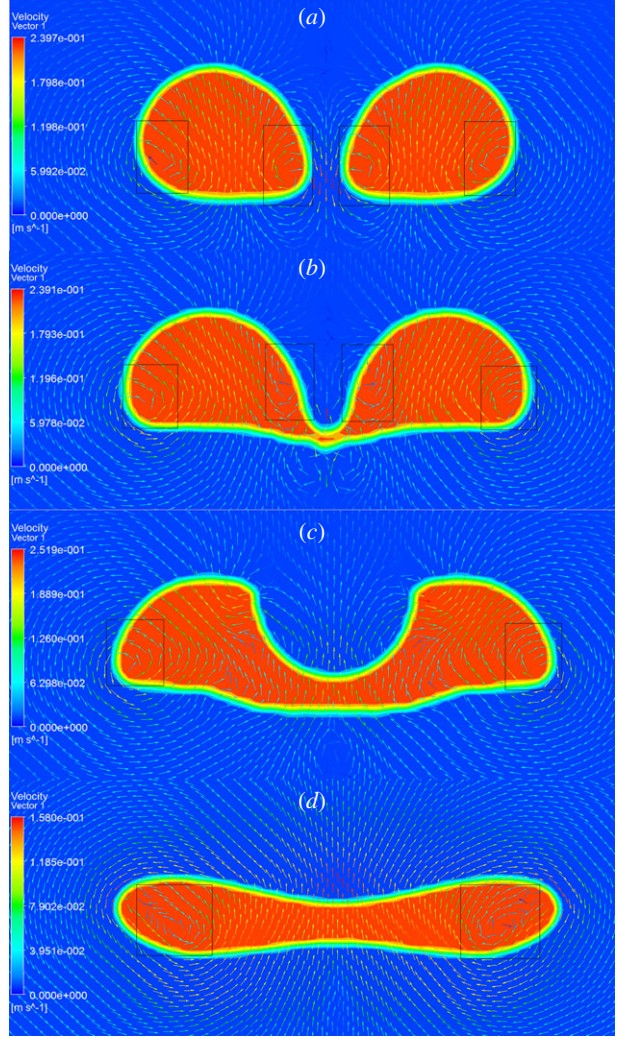

**Figure 15.** Different steps of bubble coalescence: (*a*) approach step; (*b*) touch step; (*c*) fusion step and (*d*) coalesced bubble.

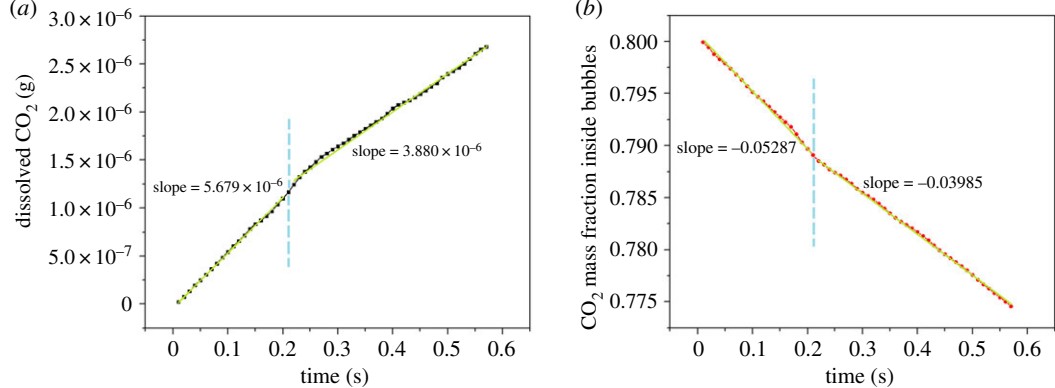

**Figure 16.** Change tendencies of (*a*) dissolved $CO_2$ and (*b*) $CO_2$ mass fraction inside 3 mm bubbles under bubble coalescence condition.

## 4. Conclusion

Based on assumptions, the VOF model was applied to simulate the behaviours of bubbles with compositions of 80 wt% $CO_2$ and 20 wt% $N_2$ in water. The solution methods can calculate accurate results, which agree with the experimental results. Along with $CO_2$ mass transfer processes, the rising

behaviour, the cutting behaviour of single bubbles and the coalescence behaviour of side-by-side bubbles were studied. The interactions of the behaviours and mass transfer were analysed based on the gas side and liquid side.

When single bubbles with different diameters rise, the path of small ones with diameters of less than 3 mm is unsteady while larger bubbles rise vertically in the domain. Through analysing the dissolved $CO_2$ in the water and $CO_2$ mass fraction inside bubbles, the results show that larger bubbles have positive impacts on the dissolution of $CO_2$ in water because of the larger interface area, but the global mass flux is lower compared with small bubbles. Besides, an accurate correlation of bubble diameter and mass flux was proposed.

Bubble cutting behaviour was divided into three steps in this paper, including initial single bubble rising step, daughter bubble rising step and coalesced bubble rising step. The change rates of dissolved $CO_2$ in water for three steps were compared and the rate of the second step is the highest. The absolute values of the slopes for $CO_2$ mass fraction inside bubbles obtained in the second step are also the highest. Moreover, the cutting of the initial large bubble and the coalescence of daughter bubbles were analysed and found to be related to the $CO_2$ mass fraction distribution compared with the bubble without mass transfer. The symmetric concentration distribution pushes the bubble vertically so that the equal-diameter daughter bubbles are formed.

Different from the coalescence of daughter bubbles, the side-by-side bubbles were added at the bottom of the domain and driven by buoyancy. The intervals between the bubbles are important for the coalescence. The critical intervals for 3, 5 and 7 mm bubbles were determined and the ratio of the critical interval to bubble diameter for smaller bubbles is higher than that for large bubbles. For the latter two bubbles, they coalesce immediately during the rise in water. However, for 3 mm bubbles, they rise away from each other initially and then move closer to contact with each other when the interval is larger than 1.5 mm. The coalescence behaviour can also be divided into three steps, which are approach step, touch step and fusion step. The changes in the flow patterns around the bubbles are caused by the coalescence of the side-by-side bubbles. Besides, compared to the mass transfer conditions, the results show that the coalescence behaviour has a negative impact on mass transfer. The concentration distribution around the bubbles is relatively homogeneous, suggesting that the mass transfer influences bubble motion slightly. The differences of critical intervals for the bubbles with and without mass transfer are not obvious, but they do exist.

Different bubble behaviours make the mass transfer efficiency different. The daughter bubbles show a better mass transfer performance than the initial large bubbles. Meanwhile, the mass transfer also has an impact on the bubble behaviours. The bubble with symmetric concentration distribution can be cut equally while the bubble without mass transfer is cut unequally. Hence, the interaction between the gas−liquid mass transfer and bubble behaviours cannot be ignored in future studies.

Data accessibility. We have described our work systematically and provided all the necessary data in the results and discussion section in the main manuscript.

Authors' contributions. X.L. and W.W. designed the study and drafted the manuscript. P.Z. and J.L. participated in data analysis. G.C. refined the manuscript. All authors gave final approval for publication.

Competing interests. We declare we have no competing interests.

Funding. This work was supported by the National Natural Science Foundation of China (grant no. 21276132); a Project of Shandong Province Higher Educational Science and Technology Program (grant no. J17KA107); the transformation project of scientific and technological achievements of Qingdao (grant no. 16-6-2-50-nsh).

Acknowledgements. We thank anonymous reviewers and editors for their insightful suggestions and careful reading.

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
