## [Reviewer comments · Royal Society Open Science]

Review History

RSOS-190136.R0 (Original submission)

Review form: Reviewer 1

Is the manuscript scientifically sound in its present form?

Yes

Are the interpretations and conclusions justified by the results?

Yes

Is the language acceptable?

Yes

Is it clear how to access all supporting data?

Yes

Do you have any ethical concerns with this paper?

No

Have you any concerns about statistical analyses in this paper?

No

Recommendation?

Major revision is needed (please make suggestions in comments)

Comments to the Author(s)

Please find the attached file (Appendix A).

Review form: Reviewer 2

Is the manuscript scientifically sound in its present form?

Yes

Are the interpretations and conclusions justified by the results?

Yes

Is the language acceptable?

No

Is it clear how to access all supporting data?

No

Do you have any ethical concerns with this paper?

No

Have you any concerns about statistical analyses in this paper?

No

Recommendation?

Major revision is needed (please make suggestions in comments)

Comments to the Author(s)

Comments to RSOS-190136

MS TITLE: Interactions between gas-liquid mass transfer and bubble behaviors

In this manuscript, the interactions between gas-liquid mass transfer and bubble behaviors (bubble breakup and coalescence) were studied with the support of VOF numerical modeling. Overall, it makes good sense for assessing the mass transfer efficiency for bubbling system and the transport of gas bubble in liquid phase.

This manuscript demonstrates well the methodology and clear objectives. However, the grammar and English used to write the manuscript needs to be double checked for errors. There are some occasions when the sentences are not clearly understood by the reader.

1. Abstract: Please provide more quantitative information of the results. Show us how the interactions between gas-liquid mass transfer and bubble behaviors happen, not only the general description.
2. Introduction: Do not just "list" the references, try to summarize and clearly show the state-of-the-art in this field.
3. Model setup: (1) Reorganize this part, try to follow the procedure: concept model, hypothesis, numerical model (include main equations, the initial and boundary condition), geometry and solving method, parameters for model building and data for validation. (2) Only buoyancy is

considered during the bubble rising, why? Show the force analysis. (3) Did the paper [32] and [33] consider "Interactions between gas-liquid mass transfer and bubble behaviors"? If not, how can you use their data for validation? If it is, what are the differences between your researches and theirs? Show more details about the two references.

4. Results and discussion: (1) you said "Bubbles with initial diameters of 4 mm, 5 mm, and 6 mm were investigated when they were cut by a wire." So add modeling information about the wire. (2) Only show the modeling result is not enough, try to show the mechanism and explain why the process occurs. For example, you say "To further study the influence of mass transfer on bubble motions, the rising process of 5 mm bubble was investigated (see Fig. 12). The white lines are axial wires. When there is no mass transfer happened, the bubble rises vertically at the very start and then it deviates from the axial wire. As a result, the bubble is cut into two unequal-diameter daughter bubbles. While as for the bubble with mass transfer, it rises along the axial wire and then is cut into two equal-diameter daughter bubbles. The comparison shows that the mass transfer cannot be ignored when studying the bubble behaviors." So, why that happen?

5. Discussion and conclusions did not clearly focus on your subject, "Interactions between gas-liquid mass transfer and bubble behaviors". In your discussion section please link your modeling results with a broader and deeper literature review, and explain how the mass transfer influence the bubble behaviors.

6. The English should be polished for publish.

In general, I recommend this manuscript "major revision".

Decision letter (RSOS-190136.R0)

12-Mar-2019

Dear Dr Chen:

Title: Interactions between gas-liquid mass transfer and bubble behaviors

Manuscript ID: RSOS-190136

The editor assigned to your manuscript has now received comments from reviewers. We would like you to revise your paper in accordance with the referee and Subject Editor suggestions which can be found below (not including confidential reports to the Editor). Please note this decision does not guarantee eventual acceptance.

Please submit your revised paper before 04-Apr-2019. Please note that the revision deadline will expire at 00.00am on this date. If we do not hear from you within this time then it will be assumed that the paper has been withdrawn. In exceptional circumstances, extensions may be possible if agreed with the Editorial Office in advance. We do not allow multiple rounds of revision so we urge you to make every effort to fully address all of the comments at this stage. If deemed necessary by the Editors, your manuscript will be sent back to one or more of the original reviewers for assessment. If the original reviewers are not available we may invite new reviewers.

Please also include the following statements alongside the other end statements. As we cannot publish your manuscript without these end statements included, if you feel that a given heading is not relevant to your paper, please nevertheless include the heading and explicitly state that it is not relevant to your work.

- Ethics statement

Please clarify whether you received ethical approval from a local ethics committee to carry out your study. If so please include details of this, including the name of the committee that gave consent in a Research Ethics section after your main text. Please also clarify whether you received informed consent for the participants to participate in the study and state this in your Research Ethics section.

OR

Please clarify whether you obtained the necessary licences and approvals from your institutional animal ethics committee before conducting your research. Please provide details of these licences and approvals in an Animal Ethics section after your main text.

OR

Please clarify whether you obtained the appropriate permissions and licences to conduct the fieldwork detailed in your study. Please provide details of these in your methods section.

- Acknowledgements

On behalf of the Subject Editor Professor Anthony Stace and the Associate Editor Professor Claire Carmalt.

RSC Associate Editor:
Comments to the Author:
(There are no comments.)

RSC Subject Editor:
 Comments to the Author:
 (There are no comments.)

Reviewers' Comments to Author:
 Reviewer: 1

Comments to the Author(s)
 Please find the attached file.

Reviewer: 2

Comments to the Author(s)
 Comments to RSOS-190136

MS TITLE: Interactions between gas-liquid mass transfer and bubble behaviors

In this manuscript, the interactions between gas-liquid mass transfer and bubble behaviors (bubble breakup and coalescence) were studied with the support of VOF numerical modeling. Overall, it makes good sense for assessing the mass transfer efficiency for bubbling system and the transport of gas bubble in liquid phase.

This manuscript demonstrates well the methodology and clear objectives. However, the grammar and English used to write the manuscript needs to be double checked for errors. There are some occasions when the sentences are not clearly understood by the reader.

1. Abstract: Please provide more quantitative information of the results. Show us how the interactions between gas-liquid mass transfer and bubble behaviors happen, not only the general description.

2. Introduction: Do not just "list" the references, try to summarize and clearly show the state-of-the-art in this field.

3. Model setup: (1) Reorganize this part, try to follow the procedure: concept model, hypothesis, numerical model (include main equations, the initial and boundary condition), geometry and solving method, parameters for model building and data for validation. (2) Only buoyancy is considered during the bubble rising, why? Show the force analysis. (3) Did the paper [32] and [33] consider "Interactions between gas-liquid mass transfer and bubble behaviors"? If not, how can you use their data for validation? If it is, what are the differences between your researches and theirs? Show more details about the two references.

4. Results and discussion: (1) you said "Bubbles with initial diameters of 4 mm, 5 mm, and 6 mm were investigated when they were cut by a wire." So add modeling information about the wire. (2) Only show the modeling result is not enough, try to show the mechanism and explain why the process occurs. For example, you say "To further study the influence of mass transfer on bubble motions, the rising process of 5 mm bubble was investigated (see Fig. 12). The white lines are axel wires. When there is no mass transfer happened, the bubble rises vertically at the very start and then it deviates from the axel wire. As a result, the bubble is cut into two unequal-diameter daughter bubbles. While as for the bubble with mass transfer, it rises along the axel wire and then is cut into two equal-diameter daughter bubbles. The comparison shows that the mass transfer cannot be ignored when studying the bubble behaviors." So, why that happen?

5. Discussion and conclusions did not clearly focus on your subject, "Interactions between gas-liquid mass transfer and bubble behaviors". In your discussion section please link your modeling results with a broader and deeper literature review, and explain how the mass transfer influence the bubble behaviors.

6. The English should be polished for publish.

In general, I recommend this manuscript "major revision".

Author's Response to Decision Letter for (RSOS-190136.R0)

See Appendix B.

Decision letter (RSOS-190136.R1)

05-Apr-2019

Dear Dr Chen:

Title: Interactions between gas-liquid mass transfer and bubble behaviors
Manuscript ID: RSOS-190136.R1

It is a pleasure to accept your manuscript in its current form for publication in Royal Society Open Science. The chemistry content of Royal Society Open Science is published in collaboration with the Royal Society of Chemistry.

On behalf of the Subject Editor Professor Anthony Stace and the Associate Editor Professor Claire Carmalt.

RSC Associate Editor
Comments to the Author:
(There are no comments.)

Reviewer(s)' Comments to Author:

Appendix A

The coalescence and cutting behavior of bubbles and mass transfer were investigated numerically. It can be published after a major revision.

1. How to control the quantity of gas from inner gas to gas-liquid interface, similarly to the mass transfer rate of CO₂ from interface to liquid?
2. Eq. 8, how to obtain $N_{\text{gas-int}}$ and $N_{\text{int-liq}}$?
3. Letters in fig. 3 are too little to present clearly.
4. The serial number of table should be rewritten in text. For example, Table 2 in page 9 line 25 should be Table 3.
5. The mass of CO₂ bubble with diameter 2 mm is about 7.5×10^{-9} kg. However, the dissolved CO₂ from bubble with diameter 2 mm into liquid is about 10^{-6} kg at 0.5 s in fig. 8. It is impossible. It is the same to other bubbles.
6. In fig. 9, please check the order of magnitude.
7. Page 13, lines 44-50, “For 3 mm bubbles with mass transfer, the ratio of 0.67 is 86.1% and 4.15 times higher than that for 5 mm bubbles and 7 mm bubbles, respectively.” It is incomprehensible. In addition, the critical interval is 2.0 mm for 3 mm bubbles, and 1.8 mm and 0.9 mm for 5 mm and 7 mm bubbles, respectively. Therefore, smaller side-by-side bubbles are more difficult to coalesce.
8. Page 14, line 7, “the 3 mm side-by-side bubbles rise on negative direction initially...” what does “negative direction” mean?
9. Nomenclature, “S” is not mentioned in text.

Appendix B

Responses to the Comments of Reviewer 1

General Comment

The coalescence and cutting behavior of bubbles and mass transfer were investigated numerically. It can be published after a major revision.

Response: We greatly appreciate your positive comments.

Comment 1: *How to control the quantity of gas from inner gas to gas-liquid interface, similarly to the mass transfer rate of CO₂ from interface to liquid?*

Response: Thank you a lot for your suggestion. In the simulation, the quantity of gas from inner gas to gas-liquid interface and the mass transfer rate of CO₂ from interface to liquid are not controlled. Based on the mass transfer mechanism, the continuous mass flux through the interface means that there is no mass accumulation. Hence, the quantity of CO₂ in the gas phase of the meshes belong to the interface was transferred to the liquid phase completely. As for the mass transfer processes in the gas phase and liquid phase, they are related to the fluid flow and species diffusion.

Comment 2: *Eq. 8, how to obtain $N_{gas-int}$ and $N_{int-liq}$?*

Response: Thank you for your advice. The mass transfer processes in the gas phase and liquid phase are related to the species diffusion and fluid flow. The $N_{gas-int}$ which represents the mass flux from gas to interface can be determined by the following equation.

$$N_{gas-int} = -D_{i,mg} \frac{\partial C_{i,g}}{\partial Z} \quad \text{Eq.R1}$$

The $N_{int-liq}$ which represents the mass flux from interface to liquid can be determined by Eq.R2.

$$N_{int-liq} = -D_{i,ml} \frac{\partial C_{i,l}}{\partial Z} \quad \text{Eq.R2}$$

There is no mass accumulation at the interface. Hence, the Eq.8 in the manuscript was revised as follow:

$$N_{gas-int} = -D_{i,mg} \frac{\partial C_{i,g}}{\partial Z} = N_{int-liq} = -D_{i,ml} \frac{\partial C_{i,l}}{\partial Z} \quad (8)$$

(Please see the highlighted parts in Paragraph 4 Page 4 in the revised manuscript)

Comment 3: Letters in fig. 3 are too little to present clearly.

Response: Thank you for your suggestion. The letters in Fig.3 of the original manuscript (Fig.4 in the revised manuscript) have been enlarged and they can be read clearly.

Figure 4. Compared results of dissolved CO₂ for different meshes: (a) 5 mm single bubble rise; (b) 3 mm side-by-side bubbles coalescence; (c) 5 mm bubble cutting.

(Please see the highlighted parts in Page 6 in the revised manuscript)

Comment 4: The serial number of table should be rewritten in text. For example, Table 2 in page 9 line 25 should be Table 3.

Response: Thank you for your suggestion. The serial number of tables has been checked and corrected.

Comment 5: The mass of CO₂ bubble with diameter 2 mm is about 7.5×10^{-9} kg. However, the dissolved CO₂ from bubble with diameter 2 mm into liquid is about 10^{-6} kg at 0.5 s in fig. 8. It is impossible. It is the same to other bubbles.

Response: Thank you for your suggestion. We simulated the CO₂ bubble rising behaviors and the mass transfer process in 2D meshes to implement the relevant studies. This simulation method is used widely in the investigations about gas-liquid mass transfer (Dani et al., International Journal of Chemical Reactor Engineering, 2006, 4:1-21; Hayashi and Tomiyama, Journal of Computational Multiphase Flows, 2011, 3:247-261; Aboulhasanzadeh et al., Chemical Engineering Science, 2012, 75:456-467; Bao et al., Chemical Engineering Science, 2015, 135:76-88). In 2D simulations, only x and y (or x and z, y and z, etc) coordinates were used. The third coordinate was set as 1 automatically which is related to the unit system. In the simulation of this study, the unit system is set as SI unit so that the length unit is meter. When calculate the dissolved CO₂ in water, volume is needed so that the deviation occurred. The size of computational domain is in the magnitude of “mm” while the third coordinate is in the magnitude of “m”. Therefore, the deviation of 10³ exists. The order of magnitude for the relevant contents in the manuscript was corrected.

Comment 6: *In fig. 9, please check the order of magnitude.*

Response: Thank you for your suggestion. The order of the magnitude in the manuscript has been checked and revised.

Comment 7: *Page 13, lines 44-50, “For 3 mm bubbles with mass transfer, the ratio of 0.67 is 86.1% and 4.15 times higher than that for 5 mm bubbles and 7 mm bubbles, respectively.” It is incomprehensible. In addition, the critical interval is 2.0 mm for 3 mm bubbles, and 1.8 mm and 0.9 mm for 5 mm and 7 mm bubbles, respectively. Therefore, smaller side-by-side bubbles are more difficult to coalesce.*

Response: Thank you for your suggestion. The sentence has been revised as follow and it is comprehensible now.

For 3 mm bubbles with mass transfer, the ratio is 0.67 while for 5 mm bubbles and 7 mm bubbles, the ratios are 0.36 and 0.13, respectively. (Please see the highlighted parts in Paragraph 2 Page14 in the revised manuscript)

The critical interval is 2.0 mm for 3 mm bubbles, and 1.8 mm and 0.9 mm for 5 mm and 7 mm bubbles, respectively. When the interval is 2.0 mm for the bubbles with different diameters, 3 mm bubble can coalesce while the 5 mm bubble and 7 mm bubble cannot. Hence, it is easy for the smaller bubbles to coalesce. To make it more comprehensible, the following contents are added in the revised manuscript.

The interval is the shortest distance between the interfaces of the two bubbles at the same horizontal line while the critical interval means that the longest interval for the bubbles to coalesce. The ratio of interval to bubble diameter reflects the difficulty of bubble coalescence. For 3 mm bubbles with mass transfer, the ratio is 0.67 while for 5 mm bubbles and 7 mm bubbles, the ratios are 0.36 and 0.13, respectively. It is clear that smaller side-by-side bubbles are easier to coalesce. For example, 3 mm bubble can coalesce while the 5 mm bubble and 7 mm bubble cannot coalesce when the interval is 2.0 mm for the bubbles. (Please see the highlighted parts in Paragraph 2 Page14 in the revised manuscript)

Comment 8: Page 14, line 7, “the 3 mm side-by-side bubbles rise on negative direction initially...” what does “negative direction” mean?

Response: Thank you for your suggestion. The “negative direction” in the sentence means that the 3 mm side-by-side bubbles rise away from each other. The sentence has been revised as follow.

Under other interval conditions, the 3 mm side-by-side bubbles rise away from each other initially and then move closer to contact with each other. (Please see the highlighted parts in Paragraph 2 Page14 in the revised manuscript)

Comment 9: Nomenclature, “S” is not mentioned in text.

Response: Thank you for your suggestion. The “S” in the nomenclature has been deleted.

Responses to the Comments of Reviewer 2

General Comment

In this manuscript, the interactions between gas-liquid mass transfer and bubble behaviors (bubble breakup and coalescence) were studied with the support of VOF numerical modeling. Overall, it makes good sense for assessing the mass transfer efficiency for bubbling system and the transport of gas bubble in liquid phase.

This manuscript demonstrates well the methodology and clear objectives. However, the grammar and English used to write the manuscript needs to be double checked for errors. There are some occasions when the sentences are not clearly understood by the reader.

Response: We greatly appreciate your encouraging and positive comments.

Comment 1: *Abstract: Please provide more quantitative information of the results. Show us how the interactions between gas-liquid mass transfer and bubble behaviors happen, not only the general description.*

Response: Thank you for your instructive advice. The abstract is revised and more quantitative information of the results is provided. The revision could reflect the interaction between gas-liquid mass transfer and bubble behavior. The contents are shown as follow.

The results show that the absolute slopes of the curves for mass fraction inside the single rising bubbles with the diameters from 3 mm to 6 mm decreases from 0.09325 to 0.02818. It means that small single bubbles have higher mass transfer efficiency. The daughter bubbles of cutting behavior and initial side-by-side bubbles of coalescence behavior also perform better than the initial large bubbles and coalesced bubbles, respectively. The bubble behaviors effect the mass transfer process. However, the latter also reacts upon the former. The critical intervals between the side-by-side bubbles decreases from 2.0 mm to 0.9 mm when the bubble diameter changes from 3 mm to 7 mm. For the coalescence behavior without mass transfer, the critical intervals are larger because there is no influence of concentration around the bubbles on the bubble motion. The coalescence of cut daughter bubbles is also influenced by the concentration. It was suggested that the interaction between the gas-liquid mass transfer and bubble behaviors cannot be ignored in the future studies.

(Please see the highlighted parts in the “Abstract” of Page 1 in the revised manuscript)

Comment 2: *Introduction: Do not just “list” the references, try to summarize and clearly show the state-of-the-art in this field.*

Response: Thank you for your suggestion. The introduction was revised and the references were summarized to clearly show the state-of-the-art in the field. The revisions are shown as follow.

The previous studies have been conducted to investigate the bubbles coalescence and breakup behaviors. The coalescence behavior can be classified as two types: in-line [5-7] and side-by-side [8, 9]. For the in-line bubbles coalescence, there are two coaxial bubbles and the upper one is called leading bubble while the other one is called trailing bubble. The drag force of the liquid acting on the bubbles [10] and the interaction between the in-line bubbles [11] were studied. The coalescence behaviors of the in-line bubbles are not only direct coalescence but also coalescence with conjunction [12]. For the side-by-side bubbles coalescence, there are two parallel bubbles rising together initially and then the interval between them decreases until they coalesce [13]. The properties of the gas and the liquid have impacts on the coalescence [14]. Just like the in-line bubbles, the coalescence behaviors of the side-by-side bubbles can also be classified as two types. One of them is direct coalescence and the other one is coalescence with bouncing [15]. Moreover, the possibility of coalescence is determined by the initial interval between the bubbles and the coalesced bubbles may also breakup if the diameter is large enough [16].

Bubble breakup behavior also usually occurs in the rising process [17-19]. It is generally recognized that the bubbles tend to breakup when the surface tension of the interface is changed greatly. There is a special bubble breakup condition called cutting, in which the bubble is cut by a wire or wire mesh set along their rising path [20, 21]. When the bubbles contact with the wire, three behaviors of cut, bypass and stick will happen. The behaviors are decided by the rising velocity and properties of gas-liquid system. When the wire mesh is set, the bubbles will contact with the center or the crossing of the meshes. Similar behaviors would occur. In our previous study [22], a wire mesh was set above the tray in the distillation column and the results shown that the mean bubble diameter decreases sharply. The above studies are of great guiding

significance to the coalescence and breakup behaviors, but the influences of the behaviors on mass transfer have not been investigated and the interaction between them is also rarely investigated.

The interaction between the bubble behaviors and the gas-liquid mass transfer for single bubble rising process has been recognized. However, the researchers paid more attention to liquid-side mass transfer, while the gas-side or coupled two-side mass transfer are rarely studied [23-26]. The liquid-side mass transfer process is influenced by the properties of the liquid such as density, surface tension and so on. Volume-of-Fluid (VOF) method is widely used in the simulation studies [27]. For the calculation of mass transfer process, a two-variable method is investigated and shows good computations for the high Schmidt and Reynolds number bubbles [28]. Direct numerical simulation (DNS) could also simulate the mass transfer process. The effect of interface contamination on mass transfer can be calculated accurately based on the method [29]. Experimental investigation about mass transfer processes is another research focus [4]. A new experiment method with some specific chromogenic methods was developed [30]. There are several advantages of the new method such as the visualization of the concentration distribution. Besides the liquid-side mass transfer, Saboni et al.[31] studied the soluble substance transportation from liquid into bubble and the concentration distribution forms in bubble based on simulation. However, the gas-side mass transfer process and the behaviors with mass transfer were rarely studied.

(Please see the highlighted parts in the “Introduction” of Page 1 in the revised manuscript)

Comment 3: *Model setup: (1) Reorganize this part, try to follow the procedure: concept model, hypothesis, numerical model (include main equations, the initial and boundary condition), geometry and solving method, parameters for model building and data for validation. (2) Only buoyancy is considered during the bubble rising, why? Show the force analysis. (3) Did the paper [32] and [33] consider “Interactions between gas-liquid mass transfer and bubble behaviors”? If not, how can you use their data for validation? If it is, what are the differences between your researches and theirs? Show more details about the two references.*

Response: Thank you for your insightful suggestions.

(1) The “Model setup” was reorganized as you suggested. Please see the relevant section.

(2) The initial bubbles are driven by buoyancy but there is also gravity force acts on the bubbles.

The force analysis was added in the revised manuscript as follow.

The buoyancy force and gravity force act on the bubble during the whole rising process (see figure 3). The total force can be calculated by Eq.9.

$$F_y = F_b - F_G \quad (9)$$

Figure 3. Force analysis for bubbles.

(Please see the highlighted parts in Page 4 and Page 5 in the revised manuscript)

(3) Paper [32] and [33] did not illustrate the interaction directly but the data are obtained by experiments. When the bubbles form and rise in the liquid, the interaction between gas-liquid mass transfer and bubble behaviors exists constitutionally though the objectives of the papers do not focus on this issue. Hence, the experimental data are obtained under the underlying premise of the interaction and they can be used to implement model validation. Besides, the calculation model about mass transfer process is usually validated by the experiment data about fluid field such as the bubble shape and rising velocity (Hayashi et al., International Journal of Multiphase Flow, 2014, 58:236-245; Bao et al., Chemical Engineering Science, 2015, 135:76-88). The simulation results can illustrate the research issue well. In this paper, the data about fluid field and concentration wakes are compared with the simulation results and the comparisons show good consistency. Therefore, the model in this paper can be validated by paper [32] and [33]. Thanks again for your insightful suggestion.

More details about the two references were added as follow.

In the work of Zahedi et al., air was injected through the orifice with 4.5 mm diameter and

water was initially filled in the domain. The air flow rate was controlled at $2.5E-7 \text{ m}^3\cdot\text{s}^{-1}$ by a pump with flow controller. The formation and rising processes of the bubble was recorded by a high speed camera with the speed of 60 frames/s. The processes were analyzed by the recorded movie files with snapshots taken every 0.016s..... Besides, the model was also validated based on the concentration wake structure of the dissolvable gas in the liquid through comparing the results obtained from Kong et al. [33]. To obtain the concentration wake structure, dual-emission dye C-SNARF-4F was selected and LIF technique was applied. The pH distribution around the bubble was captured and then the concentration distribution could be calculated by the conversion function. (Please see the highlighted parts in Paragraph 2 Page 6 in the revised manuscript)

Comment 4: *Results and discussion: (1) you said “Bubbles with initial diameters of 4 mm, 5 mm, and 6 mm were investigated when they were cut by a wire.” So add modeling information about the wire. (2) Only show the modeling result is not enough, try to show the mechanism and explain why the process occurs. For example, you say “To further study the influence of mass transfer on bubble motions, the rising process of 5 mm bubble was investigated (see Fig. 12). The white lines are axel wires. When there is no mass transfer happened, the bubble rises vertically at the very start and then it deviates from the axel wire. As a result, the bubble is cut into two unequal-diameter daughter bubbles. While as for the bubble with mass transfer, it rises along the axel wire and then is cut into two equal-diameter daughter bubbles. The comparison shows that the mass transfer cannot be ignored when studying the bubble behaviors.” So, why that happen?*

Response: Thank you for your suggestions.

(1) The modeling information about the wire was revised in the “Geometry and Solution methods” section as follow.

Under the bubble cutting condition, the tenuous wire was set at 40 mm in height from the bottom of the domain. The shape of the wire is set to square so that the grid type of the domain can be meshed as high-quality quadrilateral grids. Its side length is only 0.4 mm which is much smaller than the bubble diameters so the influence of wire shape can be ignored. The sides of the wire were set as wall boundary conditions.

(Please see the highlighted parts in Paragraph 1 Page 5 in the revised manuscript)

(2) The concentration distribution could reflect the influence of the mass transfer on bubble motion. Huang and Saito (Chemical Engineering Science, 2017, 157:182-199) verified the influence through analyzing the rising path under different conditions in the experiments. In this paper, the differences between the behaviors with and without mass transfer processes can also be analyzed by the concentration distribution. The added contents are shown as follow.

When there is no mass transfer happened, the bubble rises vertically at the very start and then it deviates from the axial wire. The deviation is caused by the fluid field and there is no mass transfer to offset the deviation. As a result, the bubble is cut into two unequal-diameter daughter bubbles. While as for the bubble with mass transfer, it rises along the axial wire and then is cut into two equal-diameter daughter bubbles. When the bubble moves close to the wire, the concentration distribution below the bubble is symmetric (see figure13c). The high concentration regions also arrange symmetrically. Hence, the bubble is pushed vertically and then cut into two equal-diameter daughter bubbles. (Please see the highlighted parts in Paragraph 1 Page 12 in the revised manuscript)

For the more detailed information about the mechanism, quantitative data of simulation and experiments need to be obtained and the analysis method also needs to be developed. This is a great research direction and we will try our best to conduct a thorough study in the future works. Thanks again for your instructive suggestion.

Comment 5: *Discussion and conclusions did not clearly focus on your subject," Interactions between gas-liquid mass transfer and bubble behaviors". In your discussion section please link your modeling results with a broader and deeper literature review, and explain how the mass transfer influence the bubble behaviors.*

Response: Thank you for your suggestion. Little literatures about the interactions between gas-liquid mass transfer and bubble behaviors for several single bubbles are really less, especially for the cutting behavior and coalescence behavior. More researches focus on the changes about fluid field or the determination of mass transfer coefficient when the above behaviors happen at the macro level (Oloms et al., Chemical Engineering Science, 2001, 56:6359-6365; Shimizu et al., Chemical Engineering Journal, 2000, 78:21-28; Wongsuchoto et al., Chemical Engineering

Journal, 2003, 92:81-90; Martin et al., Chemical Engineering Science, 2007, 62:1741-1752; Koynov et al., AIChE Journal, 2005, 51:2786-2800). We added the analysis about the interaction in the “Results and discussion” section as you suggested to highlight the subject (Please see the above responses). Besides, we revised and added the following contents in the “Conclusion” section to further illustrate the subject of this paper.

Bubble cutting behavior was divided into three steps in this paper, which are initial single bubble rising step, daughter bubbles rising step, and coalesced bubble rising step. The change rates of dissolved CO₂ in water for three steps were compared and the rate of the second step is the highest. The absolute values of the slopes for CO₂ mass fraction insides bubbles obtained in the second step are also the highest. Moreover, the cutting of the initial large bubble and the coalescence of daughter bubbles were analyzed to be related to the CO₂ mass fraction distribution compared with the bubble without mass transfer. The symmetric concentration distribution pushes the bubble vertically so that the equal-diameter daughter bubbles are formed. (Please see the highlighted parts in Paragraph 3 Page 17 in the revised manuscript)

Besides, compared the mass transfer conditions before with those after coalescing, it is found that the coalescence behavior has negative impact on mass transfer. The concentration distribution around the bubbles is relatively homogeneous, suggesting that the mass transfer influence bubble motion slightly. The differences of critical intervals for the bubbles with and without mass transfer are not obvious but they do exist.

Different bubble behaviors make the mass transfer efficiency various. The daughter bubbles shows better mass transfer performance than the initial large bubble. Meanwhile, the mass transfer also has impact on the bubble behaviors. The bubble with symmetric concentration distribution can be cut equally while the bubble without mass transfer is cut unequally. Hence, the interaction between the gas-liquid mass transfer and bubble behaviors cannot be ignored in the future studies. (Please see the highlighted parts in Page 18 in the revised manuscript)

Comment 6: *The English should be polished for publish.*

Response: Thank you for your suggestion. The English has been polished. The grammar and spelling errors have been revised. Some of the revisions are shown as follow.

Interactions between gas-liquid mass transfer and bubble behaviors were investigated to improve the understanding of the relationship between the two sides. (Please see the highlighted parts in “Abstract” of Page 1 in the revised manuscript)

The above studies are of great guiding significance to the coalescence and breakup behaviors, but the influences of the behaviors on mass transfer have not been investigated and the interaction between them is also rarely investigated. (Please see the highlighted parts in Paragraph 1 Page 2 in the revised manuscript)

However, the researchers paid more attention to liquid-side mass transfer, while the gas-side or coupled two-side mass transfer are rarely studied. (Please see the highlighted parts in Paragraph 2 Page 2 in the revised manuscript)

Based on the summary about the previous works, the coalescence and breakup behaviors for bubble rising with mass transfer need to be further investigated for improving chemical equipment performance. (Please see the highlighted parts in Paragraph 3 Page 2 in the revised manuscript)

As mentioned above, the coalescence behaviors of daughter bubbles produced from large bubbles have been analyzed, and the CO₂ mass fraction distribution around the rising bubbles influences the bubbles motion. (Please see the highlighted parts in Paragraph 1 Page 13 in the revised manuscript)

Along with CO₂ mass transfer processes, the rising behavior, the cutting behavior of single bubbles, and the coalescence behavior of side-by-side bubbles were studied. (Please see the highlighted parts in Paragraph 1 Page 17 in the revised manuscript)